# A Sensitive Impedimetric Aptasensor Based on Carbon Nanodots Modified Electrode for Detection of 17ß-Estradiol

**DOI:** 10.3390/nano10071346

**Published:** 2020-07-10

**Authors:** Mohd Hazani Mat Zaid, Jaafar Abdullah, Normazida Rozi, Aliff Aiman Mohamad Rozlan, Sharina Abu Hanifah

**Affiliations:** 1Department of Chemical Sciences, Faculty of Science and Technology, Universiti Kebangsaan Malaysia, Bangi 43600, Selangor, Malaysia; adizam_92@yahoo.com (N.R.); aliffaimanmy57@gmail.com (A.A.M.R.); 2Department of Chemistry, Faculty of Science, Universiti Putra Malaysia, Serdang 43400, Selangor Darul Ehsan, Malaysia; jafar@upm.edu.my; 3Polymer Research Centre, Faculty of Science and Technology, Universiti Kebangsaan Malaysia, Bangi 43600, Selangor, Malaysia

**Keywords:** carbon nanodots, impedance analysis, aptamer, 17ß-estradiol, aptasensor

## Abstract

A simple and sensitive aptasensor based on conductive carbon nanodots (CDs) was fabricated for the detection of 17ß-Estradiol (E2). In the present study, the hydrothermal synthesis of carbon nanodots was successfully electrodeposited on a screen-printed electrode (SPE) as a platform for immobilization of 76-mer aptamer probe. The morphology and structure of the nanomaterial were characterized by UV-visible absorption spectra, Fluorescence spectra, Transmission electron microscopy (TEM) and Fourier transform infrared spectroscopy (FTIR). Moreover, cyclic voltammetry and electrochemical impedance spectroscopy were used to investigate the electrochemical performance of the prepared electrodes. Subsequently, impedimetric (EIS) measurements were employed to investigate the relative impedances changes before and after E2 binding, which results in a linear relationship of E2 concentration in the range of 1.0 × 10^−7^ to 1.0 × 10 ^−12^ M, with a detection limit of 0.5 × 10^−12^ M. Moreover, the developed biosensor showed high selectivity toward E2 and exhibited excellent discrimination against progesterone (PRG), estriol (E3) and bisphenol A (BPA), respectively. Moreover, the average recovery rate of spiked river water samples with E2 ranged from 98.2% to 103.8%, with relative standard deviations between 1.1% and 3.8%, revealing the potential application of the present biosensor for E2 detection in water samples.

## 1. Introduction

The wave of concern over the potential health hazards associated endocrine disrupting chemicals (EDC) toward humans and wildlife began in early 1990, since the discovery of the feminized effect on male fish in rivers downstream of municipal wastewater treatment plants [1,2]. Since then, substantial research efforts have been made to study the possible adverse effects of EDCs by evaluating every type of EDC substance that may be present in the aquatic system [2]. In this group, 17ß-Estradiol (E2) has been identified as one of the emerging pollutants belonging to natural estrogen, which is also known as the most potent EDC in their class. These include estrone (E1), and estriol (E3) and its derivative ethinyl estradiol (EE2) [3,4]. Despite the fact that the reported concentration of E2 has been detected in water bodies and aquatic environments approximately in the range of 1–10 ng/L, continuous oral exposure up to the safe limit (0.3 mg/day) could induce adverse effects to a living organism in many ways, such as epigenetic changes, affecting molecular mechanisms and disrupting the reproductive system [5,6,7].

As a formidable contaminant, high-performance liquid chromatography (HPLC) [8] and gas chromatography coupled with mass spectrophotometer (GC/MS) [9] are typically used as standard instrumental analysis methods for E2 detection. This conventional analysis technique can be considered sensitive and capable of performing detection at a nanogram level; however, it is notoriously tricky to handle, especially its adaptability to carry out point-of-care detection, which often involves high operational cost and requires highly trained personnel to operate the equipment [10]. Therefore, a new approach is greatly necessary to overcome these limitations, as an alternative for the detection of E2.

Electrochemical detection is an attractive sensing platform in the field of biosensors. It avoids the requirement of expensive signal transduction equipment and provides a wide range of applications, especially for onsite monitoring [11]. Among electrochemical technique, faradic impedance biosensors showed promising approach for high sensitivity detection as a simple means, and the response was rapid [12]. Moreover, these types of sensors are a non-destructive technique that allowed them to measure surface binding events from the solution resistance continuously and provide time-dependent information on the ongoing processes of the faradaic system [13].

As any other electrochemical system, impedimetric biosensors have also been combined with various nanostructures interfaces for sensor fabrication, in order to enhance its performance [14]. Previously, surface architectures with nanomaterial became a vital prerequisite to improve electrode surface area, which can cause the increasing number of attached biological probes, and significantly influences the sensitivity of detection [15]. Carbon dots (CDs) are nanomaterials with a size below 10 nm; they were discovered coincidently during carbon nanotube multiwall purification and have opened up new potential material for the electrode modifier [16]. Previously, CDs have been applied in electrochemical sensing platforms, mainly focusing on their electrocatalytic properties toward analytes of interest [17,18] rather than electrode modifiers. Therefore, the studies on carbon dots owing a considerable potential to be used as electrode modifiers in electrochemical techniques to increase the sensitivity of the electrochemical sensor has been exploited.

In biosensor design, it is crucial to select a robust, cost-effective and straightforward recognition element to promote the success of novel biosensor development. Biosensors based on aptamers as biorecognition elements have been coined aptasensors [19] and used extensively in various analytical studies associated with drug delivery systems [20], analytical sensors [21] and therapeutic tools [22]. It demonstrated some advantages compared with traditional targeting molecules in terms of high affinity, low cost and immunogenicity, thermal stability, and reproducibility. Previously, 76-mer aptamer has gained tremendous attention in an attempt for development biosensor for E2. Kim et al. [23] reported the first study on this topic through aptamer interaction with avidin–biotin on an Au electrode chip. Since then, various sensing platforms, such as colorimetric biosensor [24], and optical readouts [25] have been used in studies using that aptamer.

In this work, we explore a novel aptasensor based on electrochemical impedance spectroscopy (EIS) derived on a conductive-carbon-dots-modified screen-printed carbon electrode (SPCE), for 17ß-Estradiol detection. The fabricated impedance electrochemical biosensor represented was the first reported biosensor using a highly conductive carbon nanodot by simple synthesis methods which used as an immobilization platform for an E2 aptamer, to improve sensitivity of detection without the necessity of any labeling strategy.

## 2. Materials and Methods

### 2.1. Chemicals

The single-strand DNA aptamer was chosen according to the prior reported literature [23]. Then, it was synthesized by IDT Inc. and used as received. The sequence of the 76-mer aptamer was labeled at the 5′-end with –NH_2_ and is given below:

NH_2_-5-GCTTCCAGCTTATTGAATTACACGCAGAGGGTA GCGGCTCTGCGCATTCAATTGCTGCGCGCTGAAGCGCG GAAGC-3

Progesterone (PRG), 17β-Estradiol (E2), estriol (E3) and bisphenol A (BPA) were purchased from Merck (Selangor, Malaysia) (Sodium monohydrogen phosphate, sodium dihydrogen phosphate, potassium ferricyanide (K_3_Fe(CN)_6_), potassium chloride (KCl), citric acid and ethylenediamine were purchased from Sigma-Aldrich (St. Louis, MO, USA). Various concentrations of E2 were obtained by dissolving standard 17β-Estradiol samples in phosphate buffer (pH 7.2).

### 2.2. Apparatus and Instrumentation

Screen-printed carbon electrodes (SPCEs) were purchased from Rapid lab Sdn Bhd (Selangor, Malaysia). The electrodes incorporate a conventional three-electrode configuration with adaptation to a microwell, to enable electrolyte control efficiently. All electrochemical measurements were carried out by using Autolab PGSTAT204 (Utrecht, Netherlands), controlled by the computer. Cyclic voltammetry and electrochemical impedance spectroscopy (EIS) studies were performed in 5 mmol Fe(CN)_6_^3−^/Fe(CN)_6_^4−^ a solution containing 0.1 mol L^−1^ KCl solution, as probe mediator. Impedance measurements were recorded between 0.1 MHz and 0.1 Hz at a sinusoidal voltage perturbation of 5 mV amplitude. The Nyquist plot was fitted and analyzed by using the Randles equivalent circuit, which consisted of the solution or electrolyte resistance (R_S_) connected in series to the parallel combination of the constant phase element (CPE) and charge transfer resistance (R_P_) in series with Warburg impedance (W) [26].

### 2.3. Synthesis of Carbon Dots

CDs were synthesized via a hydrothermal route with citric acid and EDA, as described by Zhu et al. [27]. Firstly, citric acid (3.0 g) and ethylenediamine (1875 µL) were dissolved in 30 mL of distilled water. Then the solution was transferred into a 500 mL round-bottom flask and heated with a temperature of 150 °C for 5 h. The product was subjected to dialysis against the ddH_2_O (retained molecular weight: 3500 Da) to obtain the CDs. CDs powder was obtained by evaporation, re-dispersed in deionized water and stored at 4 °C for further use.

### 2.4. Electrode Modification

In this study, screen-printed electrodes (SPCEs) were fabricated with microwell, which was used to improve detection efficiency, and were modified with CDs, through the electrodeposition technique. Prior to electrode modification, pretreatment of the SPCE is necessary to activate the carbon electrode surface. SPCE was pretreated by the cyclic voltammetry (CV) for ten cycles in the potential range of +0.1 to +0.7 V, and the scan rate of 50 mV/s in 0.1 M phosphate buffer solution (PBS). To deposit CDs on the SPCE, CDs solution (1 mg/L) was mixed with 0.1 M KCl solution at a volume ratio of 4:1 and dropped onto microwell fabricated SPCE. Then, CV was conducted in the potential ranges of −0.2 to 0.2 V, for 50 cycles, at scan rate of 50 mV/S.

### 2.5. Aptasensor Preparation

Firstly, the modified electrode was treated with the homobifunctional cross-linker glutaraldehyde for 1 h. Afterward, the prepared electrodes were then washed with 0.1 M phosphate buffer (PBS) pH 7.0 three times (5 s each), to remove uncrosslinked glutaraldehyde. Subsequently, 3 µL of amine-functionalized aptamer was dropped onto the prepared electrode for 2 h, at room temperature. The electrode surface was again washed, using phosphate buffer (PBS) (pH 7), for 5 s, to remove any unbound aptamers on the surface electrode. Prior to E2 incubation, the unreacted aldehyde functional groups on the modified SPCE should be blocked by reacting them with the sodium borohydride (Na_2_BH_4_) [28]. Then, 3 µL of E2 was incubated on aptamer/CDs/SPCE for 15 min, at room temperature. After incubation, the electrode rinses thoroughly with PBS, to remove the unbound E2, before performing detection. After the detection, the relative impedance changes (∆Rct %) were calculated based on Rct_o_-Rct_b_ divided by Rct_b_, where Rct_0_ is the impedance of the aptamer without the presence E2 binding, while Rct_b_ is the impedance of the aptamer with a different concentration of 17β–estradiol. Scheme 1 shows the process flow for the fabrication of impedimetric aptasensor for the determination of 17β-Estradiol.

## 3. Results

### 3.1. Material Characterization

The prepared CDs were thoroughly characterized by using UV/Vis, fluorescence spectrophotometer, transmission electron microscopy (TEM) and Fourier-transform infrared spectroscopy (FTIR), respectively. In Figure 1A, the UV/Vis and fluorescence spectra of CDs was observed when excited at wavelength 340 nm and are attributed to the n–π* transitions of C=O [29], resulting in a fluorescence emission peak at wavelength 443 nm [30]. The synthesized CDs were further investigated by TEM, as shown in Figure 1B. It can be seen that the size of an individual CD consists of non-uniform fine particles, with a size range of 3–5 nm, which is most often reported as the average diameter in the previous reported works [31,32]. Moreover, Figure 1C shows FTIR analysis of CDs displays the presence of OH and NH functional groups, with the peak observed in the region of 3000–3600 cm^−1^. Meanwhile, the sharp peak at 1554 cm^−1^ corresponds to N–O stretching, and the peak at 1380 cm^−1^ represents C–H bending. Moreover, the peaks at 1259 cm^−1^ represent C–N stretching and 1070 cm^−1^ from C–O, respectively.

### 3.2. Electrode Modification and Characterization

Electrode modification plays a crucial key to produce stable and reproducible results to enhance the active surface area of the electrode. It is necessary to use one particular technique to fix the inconsistency of electrode performance and electrically conductive properties. Instead of drop-casting means, the electrodeposition technique was chosen as a method for modifying the SPCE with CDs in this study. The electrodeposition of CDs on SPCE was carried out by setting up the potential in the range from −0.2 to 0.2 V in the presence of 0.1 M KCl solution as an electrolyte solution. As shown in Figure 2A, the cyclic voltammogram of CDs/SPCE shows a small anodic potential at 0.08 V (peak i), and a cathodic peak at −0.12 V (peak ii) can be observed after the electrodeposition process. The appearance of these peaks is probably due to the existence to oxidation and reduction of citric acid and ethylenediamine. This typical peak could also be seen in cyclic voltammograms (CVs) for the ascorbic acid, which is derivate to citric acid in the previous study.

Furthermore, the changes in morphology that occurred after CV electrodeposition were characterized by field emission scanning electron microscopy (FESEM, CRIM, UKM, Selangor, Malysia). It can be seen that the modification SPCE exhibited a more packed and agglomerated nanostructure (Figure 2B) compared to bare SPCE (Appendix A), indicating the success of the use of deposition method. In order to monitor conductivity on the modified electrode, typical cyclic voltammetry (CV) was carried out in the presence of 5 mM Ferrocyanide/ferricyanide (Fe(CN)_6_^3−/4−)^ redox couple containing 0.1 M KCl, as shown in Figure 2C. In this study, bare SPCE (Figure 2C, curve a) exhibited irreversible behavior redox peak, which indicates a slow electron transfer process than CDs/SPCE, which demonstrated a reversible and well-defined system due to fast electron transfer process [33]. On the other hand, the modified SPCE (Figure 2C, curve b) displayed a high redox peak current with less peak-to-peak separation (∆Ep), indicating excellent conductivity of CDs through its accomplishment of diffusional control by [Fe(CN)_6_]^3^^−/4^^−^ ion on the surface of the electrode [34]. Subsequently, electrochemical impedance analysis was carried out, which acts as a complement to CV results, as shown in Figure 2D. As a result, the Nyquist plot of bare SPCE (Figure 2D, curve a) shows a large semicircle domain (curve a) with a charge transfer resistance (Rct) value of about 254 kΩ. Meanwhile, CDs/SPCE (Figure 2D, curve b) shows a much lower Rct value of about 27.1 kΩ, which suggested that acceleration electron transfer between the electrochemical probe [Fe(CN)_6_]^3−/4−^ and electrode surface has increased due to the enhancement electrode surface area. These results can be calculated by using the Randles–Sevcik equation [35], where a CV experiment was performed in 5.0 mM Fe (CN)_6_
^4−/3−^ solution containing 0.1 M KCl at different scan rates (v) (Appendix A). As a result, the electroactive surface area (A) of the bare SPCE and the CDs/SPCE electrode was determined to be 0.019 cm^2^ (Appendix A) and 0.054 cm^2^ (Appendix A), respectively, which translates to an approximately three-times improvement compared to bare SPCE (Appendix A).

### 3.3. Design and Construction the Developed Aptasensor

In order to understand surface organization at each stage of electrode modification, EIS measurements were recorded in the [Fe(CN)_6_]^3−/4−^ at four different phases during sensor development: (1) CDs/SPCE; (2) CDs/SPCE/glutaraldehyde; (3) CDs/SPCE/ glutaraldehyde/Aptamer; and (4) CDs/SPCE/ glutaraldehyde/Aptamer/E2, respectively. As can be seen, the small diameter of the semicircles of the Nyquist plots was obtained with CDs/SPCE, as shown in Figure 3A curve a. However, incubation with the glutaraldehyde showed a dramatic increase in Rct about 73% (Figure 3A, curve b) and indicates that the formation of linker on the modified electrode caused current passivation on the electrode surface and impede the electron movement. Moreover, on subsequent immobilization with the aptamer results in more increases of Rct value corresponding with a large curve diameter (Figure 3A, curve c) due to electrostatic repulsion force between negatively charged aptamers probe and redox couple [Fe(CN)_6_]^3−/4−^ anions [36]. Subsequently, a drastic change of EIS signal showed 67% decrement of Rct can be seen after incubation of E2 (Figure 3A, Curve d), which indicates an immobilized aptamer successful bind specifically to the E2. These phenomena can be explained by the folded G-quadruplex configuration of E2-aptamers that forces E2 into proximity of the aptasensing CDs/SPCE/glutaraldehyde/Aptamer interface and reduces the electrostatic repulsion between aptamer and [Fe(CN)_6_]^3−/4−^, leading to improved electron-transfer efficiency. This result is in agreement with previously reported E2 aptasensing, where the result showed that an increase in E2 concentration induced an increase in the current response upon the binding E2 toward the aptamer [37].

Further studies have been carried out on aptamer surface density on the modified electrode (CDs/SPCE) and bare SPCE, respectively. This study was done after we realized a major change in Rct value after E2 incubation was probably due to the broader coverage of aptamer on the modified electrode. Briefly, this approach is based on the chronocoulometric method reported by Steel et al. (1998) [38], with the assumption that cationic redox molecule hexaammine ruthenium(III)chloride [Ru(NH_3_)_6_]^3+^ (100µM) would bind with the anionic aptamer phosphate backbone in a given time period. In order to compare the density of aptamer between bare SPCE and modified electrode, the redox charges of [Ru(NH_3_)_6_]^3^ were taken in the absence and presence of 100 µM [Ru(NH_3_)_6_]^3+^, respectively, using chronocoulometric. The charge (Q) as a function of time (t) in chronocoulometric is given by the integrated Cottrell equation.
Q = 2nFAD_o_^1/2^C_o_t^1/2^ ⁄π^1/2^ + Q_dl_ + nFAГ_o_(1)
where n is the number of electrons per molecule for reduction (n = 1), F is the Faraday constant (96485 C/eq), A is the (cm_2_/s), D_o_ the diffusion coefficient (cm^2^/s), C_o_ the bulk concentration of Ru(NH_3_)_6_^3+^ (mol/cm^3^), Q_dl_ the capacitive charge (C) and nFAΓ_o_ the charge from the reduction of Γ_0_, the amount of surface confined redox marker (mol/cm^2^). Subsequently, the intercept difference of Q~t^1/2^ between curve b and curve a represents the aptamer on the bare SPCE, while curve d and curve c represent the aptamer on the modified electrode. It should be noted that the abovementioned equation assumes that the surface density is determined from the saturated surface excess from [Ru(NH_3_)_6_]^3+^:ΓDNA = Γ_0_(z/m)NA(2)
where ΓDNA is the DNA probe surface density (mol/cm^2^), m is the number of bases in DNA probe (76 base pair), z is the charge on redox marker and NA is Avogadro’s number [38]. As a result, the surface density obtained was 8.3 × 10^12^ molecule/cm^2^ compared to the bare electrode’s, which was 1.12 × 10^10^ molecule/cm_2_ with the same immobilization procedure. This result shows that the higher packing aptamer density on the modified electrode probably occurs due to a large effective surface area of the modified electrode where a large amount of aptamer can be attach compared to bare SPCE, which has a less-effective surface area.

### 3.4. Analytical Performance of the Developed Aptabiosensor

Figure 4A shows that the EIS responses for E2 at different concentrations were recorded in 5 mM ferro/ferricyanide [(Fe(CN)_6_] ^4-/3^-) containing 0.1 M KCl. From Figure 4A, it can be seen that the Rct value decreased with the increasing of E2 concentration after the binding occurred between aptamer and E2. This phenomenon can be explained by considering the reduced range of electrostatic repulsion between aptamer and [Fe(CN)_6]_^3-/4-^, due to folded E2-Aptamer and causing increased access in redox species on the electrode surface [39]. Subsequently, the limit detection of aptasensor was calculated to be 0.5 × 10^−12^ M, using the 3σ/slope ratio (in which σ is the standard deviation of zero-dose response), based on the curve plotted between relative impedance change and aptamer concentration in the range of 1.5 × 10^−9^–1 × 10^−12^ M (defined S/N = 3), which can be described by the linear regression equation of 1.331× + 26.57, with a correlation coefficient of 0.993. The obtained LOD in this study is considered low and sensitive compared to some existing aptabiosensors in the previous literature, using the 76-mer aptamer (Table 1). Overall, our fabricated aptasensor offers a few advantages compared to the previously reported strategy for the determination of E2 in term of sensitivity, simple and possible-to-use large-scale production. Even though several research studies have reported sensor-based aptamers with different lengths that have shown higher LOD [40], those aptamers are not suitable for onsite application, since glass carbon electrode (GCE) was used as a working electrode, instead of disposable SPCE, which was used in this study. Meanwhile, the latest reported molecularly imprinted polymer (MIP)-based biosensors for E2 [41], owing to their advantages in terms of sensitivity, seem to have become a promising substitute for the aptamer in the development of E2 biosensors.

### 3.5. Specificity and Stability of the Developed Aptasensor

Three different compounds commonly found in water bodies have been studied as interference to the target 17β-estradiol and further tested with the aptamer on CDs/SPCE. The method of studies was adapted from Ke et al., 2014 [46], where the interference chemicals were added, along with E2, into the same vile, which contained 1 × 10^−9^ M of E2. In this study, each bar was plotted by dividing the value of the impedance corresponding to 1 nM E2 with each impedance corresponding to disrupting chemical. Thus, the changes occurred less than 5%, indicating that there was no interference effect resulting from unknown species. This can be seen in Figure 5A, where the mixture between E2 and three different interference (BPA), (E3) and (PRG) has shown 3.7%, 4.8% and 4.7% relative response changes for each bar respectively. Besides that, three mixtures of interference chemicals were also added with E2, and the relative impedance response change was less than 15%, indicating that nonspecific binding occurred, which also shows excellent specificity of the developed aptasensor in the complex water system. In addition, to evaluate the stability of the developed aptasensor of CDs/SPCE, the developed aptasensenor was independently prepared and stored at 4 °C for 30 days before use, to determine 1 nM of E2. As a result, the relative impedance changes of CDs/SPCE decreased by only 5% after two weeks of storage than the original response (Figure 5B). Moreover, after 30 days of storage, the relative response decreased to 5.3% of the original response, demonstrating that the aptasensor exhibits sufficient stability for the detection of E2.

### 3.6. Analytical Application to Real Samples

River water was used to further investigate the practicability of the developed biosensor in the real environment sample. The river water samples used in this study were collected from Langat river (Malaysia), located in Bangi Selangor. The collected samples were filtered with a 0.45 µm filter membrane prior to use, to remove suspended particles. Subsequently, the samples were spiked by three different concentrations of E2, in order to determine the recoveries of E2 in river water. As shown in Table 2, the improvements (n = 3) of the measured samples by the aptsensor showed the recovery obtained was between 92.3% and 101.2% and the relative standard deviations (RSD) (n = 3) were in the range of 0.9% and 1.5%. These results indicate that the developed aptasensor can resist the interference of the complex matrices and can be applied to determine E2 in the different aquatic systems where E2 has the potential to leach into food and water supplies and humans are widely exposed to it.

## 4. Conclusions

In this work, we have demonstrated a successful design and preparation of an impedimetric aptasensor modified electrode for the detection 17β-Estradiol (E2), using conductive carbon nanodots synthesized by a simple hydrothermal method. The developed aptasensor showed excellent analytical performance for E2 detection, with high sensitivity, specificity and reproducibility. A low detection limit of 0.5 pM was obtained, which was comparable or even lower than the previous reports, by using similar a 76 mer aptamer targeted 17β-estradiol. Moreover, the aptasensor was successfully applied to detect the real samples, and satisfactory recoveries were obtained.

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
