# Peer review of "A Sensitive Impedimetric Aptasensor Based on Carbon Nanodots Modified Electrode for Detection of 17ß-Estradiol"

_nanomaterials, 2020, doi:10.3390/nano10071346_

Round 1

Reviewer 1 Report

The work “A sensitive impedimetric aptasensor based on carbon 2 nanodots modified electrode for detection of 17ß-3 Estradiol” by Mohd Hazani Mat Zaid et al presents label-free detection of 17ß-Estradiol using SPCE modified with conductive carbon nanodot. The topic is interesting and the work is well presented.  In order to be accepted at MDPI Nanomaterials, mandatory revisions are required. In particular:

·        It should be clearly stated what the levels of 17ß-Estradiol (E2) considered to be a contaminant are.

·        In line 102-103 and 169 authors should write properly the chemical formulas of ferrocyanide and ferricyanide. All the units all over the text should be corrected and written properly.

·        Authors should tell the concentration of the carbon nanodot and the amount which was drop cast on the electrode before to electrodeposited them.

·        In scheme 1, the inset graph is not readable. Authors should improve this. In general all pictures are blurred and not well presented and should be drastically improved including Scheme 1, Fig. 1, 2, 3, 4 and 5. In fig 2 for examples the caption of the SEM is not readable at all! Labels should be added everywhere, text width should be similar all over the figures! Abbreviations should be included. 

·        In line 144-148 the authors mention that the presence of the carboxylic group is observed due to FTIR analysis. In the FTIR plot (Fig. 1 C) in the range 1760-1665 cm-1 there is no peak, which indicates the absence of the carbonyl group. From the FTIR plot it seems that in carbon nanodot you have hydroxylic groups rather than carboxylic groups. The authors have to consider this change and explain it thoughtfully.

·        In figure 2 (B) the legend is nor readable. Authors should improve it.

·        In line 294-295 the authors explain very good sensor stability with a decrease to 5.3% of the original response after 30 days. However, picture 5 (B) presenting sensor stability is not clear. The author should change the plot to make it understandable and clear.

Author Response

Thank you for your comment and the opportunity to revise our paper on ‘A sensitive impedimetric aptasensor based on carbon nanodots modified electrode for detection of 17ß-Estradiol.  The suggestions offered by the reviewers have been immensely helpful, and we also appreciate your insightful comments on revising entire aspects of the paper.

I have included the reviewer comments immediately after this letter and responded to them individually, indicating exactly how we addressed each concern or problem and describing the changes we have made. The revisions have been approved by all four authors and I have again been chosen as the corresponding author. The changes are marked in red in the paper as you requested, and the revised manuscript is attached to this email message.

Response to Reviewer 1

The work “A sensitive impedimetric aptasensor based on carbon 2 nanodots modified electrode for detection of 17ß-3 Estradiol” by Mohd Hazani Mat Zaid et al. presents label-free detection of 17ß-Estradiol using SPCE modified with conductive carbon nanodot. The topic is interesting and the work is well presented.  In order to be accepted at MDPI Nanomaterials, mandatory revisions are required. In particular:

  1. It should be clearly stated what the levels of 17ß-Estradiol (E2) considered to be a contaminant are.

  Response: Thank you for the suggestion, we added new information regarding the level of 17ß-      Estradiol (E2) problems for the aquatic organisms and animals in many aquatic systems as suggested  by the reviewer. Please see the changes in 39.

  1. In line 102-103 and 169 authors should write properly the chemical formulas of ferrocyanide and ferricyanide. All the units all over the text should be corrected and written correctly.

Response: thanks for the comment, we had fixed the mistake, and the entire text should be    corrected now.

  1. Authors should tell the concentration of the carbon nanodot and the amount which was drop cast on the electrode before to electrodeposited them.

Response: Thank you for pointing this out, we noticed that shortcoming. We have added a  new information on CDs concentration. Please see changes in line 122.

  1. In scheme 1, the inset graph is not readable. Authors should improve this. In general, all pictures are blurred and not well presented and should be drastically improved, including Scheme 1, Fig. 1, 2, 3, 4 and 5. In fig 2 for examples, the caption of the SEM is not readable at all! Labels should be added everywhere, text width should be similar all over the figures! Abbreviations should be included.

Response: Thank you for this suggestion. We had improved the Figure to make more precise and well presented as a suggestion by the reviewer.

  1. In line 144-148 the authors mention that the presence of the carboxylic group is observed due to FTIR analysis. In the FTIR plot (Fig. 1 C) in the range 1760-1665 cm-1 there is no peak, which indicates the absence of the carbonyl group. From the FTIR plot it seems that in carbon nanodot you have hydroxylic groups rather than carboxylic groups. The authors have to consider this change and explain it thoughtfully.

Response: Thank you for your  keen observation. We agree with your explanation. Therefore, we have change the FTIR spectra as shown in Fig 1c to shows the existing functional group in this study

  1. In figure 2 (B) the legend is nor readable. Authors should improve it.

Response: We agree with this and have incorporated your suggestion throughout the manuscript

  1. In line 294-295 the authors explain very good sensor stability with a decrease to 5.3% of the original response after 30 days. However, picture 5 (B) presenting sensor stability is not clear. The author should change the plot to make it understandable and clear.

Response: Thank you for the suggestion. However there is mistakes on stability day. It supposed 28 days instead of 30 days.  As suggested by the reviewer, we have changed a new plot of stability graph.

Reviewer 2 Report

The manuscript « A sensitive impedimetric aptasensor based on carbon nanodots modified electrode for detection of 17ß-Estradiol” by Mat Zaid et al., presents development of an electrochemical aptasensor for 17beta-estradol.

  1. The manuscript need to be carefully verified for spelling and grammar errors.
  2. Abstract has to be rewritten!

For instance : … In the present study, the synthesis carbon nanodots were successfully being electrodeposited on screen printed electrode (SPE) has been developed as a platform for immobilization of 76-mer aptamer as a detection probe.

or

Subsequently, impedimetric EIS) measurements were employed to investigate the changes that occurred during the modification showed a dramatic shift in electron transfer resistance value before and after E2 addition using impedance analysis….

  1. The whole manuscript needs English editing.
  2. During functionalization electrodes were washed to remove unbound glutaraldehyde and aptamer for 5 sec. Why such a short time was used?
  3. LOD was calculated using equation (3). However, this equation cannot be applied for a logarithmic curve. Another method should be used to calculate LOD using data from Fig 4. Otherwise, several low concentrations of E2 have to be measured and the calculation should be performed starting from a linear curve.

Author Response

Dear Editor/ Reviewer

Thank you for your comment and the opportunity to revise our paper on ‘A sensitive impedimetric aptasensor based on carbon nanodots modified electrode for detection of 17ß-Estradiol.  The suggestions offered by the reviewers have been immensely helpful, and we also appreciate your insightful comments on revising entire aspects of the paper.

I have included the reviewer comments immediately after this letter and responded to them individually, indicating exactly how we addressed each concern or problem and describing the changes we have made. The revisions have been approved by all four authors and I have again been chosen as the corresponding author. The changes are marked in red in the paper as you requested, and the revised manuscript is attached to this email message.

Reviewer 2

The manuscript « A sensitive impedimetric aptasensor based on carbon nanodots modified electrode for detection of 17ß-Estradiol” by Mat Zaid et al., presents development of an electrochemical aptasensor for 17beta-estradol.

  1. The manuscript need to be carefully verified for spelling and grammar errors.

Response: Thank you for your kind suggestions. Grammatical errors have been checked.

  1. Abstract has to be rewritten! For instance : … In the present study, the synthesis carbon nanodots were successfully being electrodeposited on screen printed electrode (SPE) has been developed as a platform for immobilization of 76-mer aptamer as a detection probe. Or Subsequently, impedimetric EIS) measurements were employed to investigate the changes that occurred during the modification showed a dramatic shift in electron transfer resistance value before and after E2 addition using impedance analysis….

Response:  Thank you for your comment.  We agree that abstract has to be rewritten. Please see the changes have been made in line 17.

  1. The whole manuscript needs English editing.

Response: Thanks for your suggestions. These types of errors have been carefully revised, and the language has been improved

  1. During functionalization electrodes were washed to remove unbound glutaraldehyde and aptamer for 5 sec. Why was such a short time used?

Response: Thanks for pointing this out. After working different washing time, we don’t see any significant different on Rct results. Therefore we decide to use 5 second ± 0.30 which we think it enough to remove unbound glutaraldehyde and aptamer

.

  1. LOD was calculated using equation (3). However, this equation cannot be applied for a logarithmic curve. Another method should be used to calculate LOD using data from Fig 4. Otherwise, several low concentrations of E2 have to be measured and the calculation should be performed starting from a linear curve.

Response: Thanks for the question. We are  not using equation 3 to calculate the LOD of the developed biosensor. However, we believe that an error in our explanation may raise this question. Therefore, we have changed the explanation in the text to make it understandable. Please see the changes in line 262.

Reviewer 3 Report

In this manuscript, the authors have reported an experimental study of using conductive carbon dots for the sensitivity improvement of electrical chemical aptasensors with 17ß-Estradiol (E2) as targeted molecules. The carbon dots were immobilized by an electro-deposition process on screen printed electrodes with 76-mer aptamers as capture probe. The Electrochemical Impedance measurements were employed to monitor the shift of electron transfer resistance before and after the binding of target 17ß-Estradiol (E2). A linear signal changes was obtained for E2 concentrations ranging from of 1.0 x 10-8 to 1.0 x -12. The selectivity experiments were also carried out with control chemicals of progesterone (PRG), estriol (E3) and bisphenol A (BPA). This work looks interesting and promising. I recommend the paper for publishing in this journal after the below concerns have been addressed:

Comments:

1. In Page 4, there is no figure caption for the schematic figure 1 for the illustration. Also, there is no “Scheme 1” mentioned in the context of the manuscript.

2. In Page 5, “Fig 1: (A) UV-Vis absorption spectrum and fluorescence spectrum of CDs; (B) TEM images of CDs; (C) FTIR spectra of CDs.” The authors have described that “Firstly, citric acid (3.0 g) and ethylenediamine (1875 μ L) were dissolved in 30 ml of distilled water. Then the solution was transferred into a 500 ml round bottom flask and heated with a temperature of 150oC for 5 h. The product was subjected to dialysis against the ddH2O (retained molecular weight: 3500 Da) in order to obtain the CDs.” However, in Figure 1b, the sizes of the carbon dots are shown that they are not uniform. And most of them are not in a spherical shape. The authors should explain this point and provide the measurement data of the zeta potential and size distribution of the carbon dots.

3. In Page 9, Figure 4. “(A) Impedance spectra (Nyquist plots) of the developed CDs/SPCE aptasensor for different concentrations of E2: (B) Linear plots between log Concentration of E2 and relative impedance change (%)” The authors have shown the results for E2 with a low concentration of 10-12 mol/L. However, they did not mention the sample volume that were used for the detection, and also after the binding process, when they unbounded molecules are flushed away, how does the signal changes? The signal will probably decrease to a lower level.

4. In Page 11, “Fig 5: (A) Specificity of the EIS aptamer biosensor toward E2; (B) Stability studies Analytical application to real samples using 1nM of E2.” The authors have shown the mixture between E2 and three different interference (BPA), (E3) and (PRG) of 3.7%, 4.8% 4.7% relative responses change for each bar respectively. Besides that, three mixture of interferences chemical also was added with E2 and the relative impedance response changed obtains is less than 15%.” However, the error bars for E3 and PRG are very large, and the lowest value is close to the E2 results. Thus, these data are not strongly supporting the selectivity properties of the CDs-based sensors for E2 only.

Author Response

Dear Editor/ Reviewer

Thank you for your comment and the opportunity to revise our paper on ‘A sensitive impedimetric aptasensor based on carbon nanodots modified electrode for detection of 17ß-Estradiol.  The suggestions offered by the reviewers have been immensely helpful, and we also appreciate your insightful comments on revising entire aspects of the paper.

I have included the reviewer comments immediately after this letter and responded to them individually, indicating exactly how we addressed each concern or problem and describing the changes we have made. The revisions have been approved by all four authors and I have again been chosen as the corresponding author. The changes are marked in red in the paper as you requested, and the revised manuscript is attached to this email message.

Response to Reviewer 3

In this manuscript, the authors have reported an experimental study of using conductive carbon dots for the sensitivity improvement of electrical chemical aptasensors with 17ß-Estradiol (E2) as targeted molecules. The carbon dots were immobilized by an electro-deposition process on screen printed electrodes with 76-mer aptamers as capture probe. The Electrochemical Impedance measurements were employed to monitor the shift of electron transfer resistance before and after the binding of target 17ß-Estradiol (E2). A linear signal changes was obtained for E2 concentrations ranging from of 1.0 x 10-8 to 1.0 x -12. The selectivity experiments were also carried out with control chemicals of progesterone (PRG), estriol (E3) and bisphenol A (BPA). This work looks interesting and promising. I recommend the paper for publishing in this journal after the below concerns have been addressed:

Comments:

  1. In Page 4, there is no figure caption for the schematic figure 1 for the illustration. Also, there is no “Scheme 1” mentioned in the context of the manuscript.

Response: Thank you for pointing this out. We agree with this.  Therefore, we have improved the  journal text  according to your suggestion. (Please the changes make in line 129) 

  1. In Page 5, “Fig 1: (A) UV-Vis absorption spectrum and fluorescence spectrum of CDs; (B) TEM images of CDs; (C) FTIR spectra of CDs.” The authors have described that “Firstly, citric acid (3.0 g) and ethylenediamine (1875 μ L) were dissolved in 30 ml of distilled water. Then the solution was transferred into a 500 ml round bottom flask and heated with a temperature of 150oC for 5 h. The product was subjected to dialysis against the ddH2O (retained molecular weight: 3500 Da) in order to obtain the CDs.” However, in Figure 1b, the sizes of the carbon dots are shown that they are not uniform. And most of them are not in a spherical shape. The authors should explain this point and provide the measurement data of the zeta potential and size distribution of the carbon dots.

Response: Thank you for pointing this out.  The obtained CDs in this study is following the previous preparation methods by Zhu et al. [27]. where most of the results obtained show similar results as we got in this study with an average diameter of 3 nm . However, the captured carbon dots were practically  may not in spherical in shape. Therefore,  we agree with this and have incorporated your suggestion throughout the manuscript (In line 150).  Due to some limitation and current situation in our country, we were not able to carry out the zeta potential anaysis to support the size distribution that you suggested

  1. In Page 9, Figure 4. “(A) Impedance spectra (Nyquist plots) of the developed CDs/SPCE aptasensor for different concentrations of E2: (B) Linear plots between log Concentration of E2 and relative impedance change (%)” The authors have shown the results for E2 with a low concentration of 10-12 mol/L. However, they did not mention the sample volume that was used for the detection, and also after the binding process, when the unbounded molecules are flushed away, how does the signal changes? The signal will probably decrease to a lower level.

Response: Thank you for your keen observation. We noticed the misinformation on part of sample volume. Therefore, we added information regarding the volume sample in the text. Please see the changes in line 130.  About your second question we needs to clarify  that the measurement with EIS only being carried only after washing step, as we mention in line 132 where the electrode rinses thoroughly with PBS to remove the abound E2 before performing detection. Therefore, the signal change after  unbounded molecules are flushed away can be seen In fig 3A (curve c).  

  1. In Page 11, “Fig 5: (A) Specificity of the EIS aptamer biosensor toward E2; (B) Stability studies Analytical application to real samples using 1nM of E2.” The authors have shown the mixture between E2 and three different interference (BPA), (E3) and (PRG) of 7%, 4.8% 4.7% relative responses change for each bar respectively. Besides that, three mixture of interferences chemical also was added with E2 and the relative impedance response changed obtains is less than 15%.” However, the error bars for E3 and PRG are very large, and the lowest value is close to the E2 results. Thus, these data are not strongly supporting the selectivity properties of the CDs-based sensors for E2 only.

Response: Thank you for the question. The value of 3.7%, 4.8% 4.7% inside the graph is not represent RSD value by direct value of Rct. The each bar we got in this study is based on Relative Response is obtained by setting the calculated impedance of the aptasensor after incubated with 1 x 10-7 M of E2 as 100% firstly, then use the impedance corresponding to E2 dividing each impedance corresponding to disrupting chemical (BPA), (E3) and (PRG). The concentration of adding disrupting chemical is 100 times as that of E2, the Relative Responses change less than 5%, indicating has no interference with the E2 aptasensor.

Round 2

Reviewer 1 Report

The authors have appropriately revised the manuscript. The quality of the figures (especially figure 1) needs however to be further improved.

Author Response

We thank you for your careful reading of the manuscript and helpful comments and suggestions. We have made revisions according to your comments and suggestions, as described below

Reviewer 1

The authors have appropriately revised the manuscript. The quality of the figures (especially figure 1) needs however to be further improved.

Response:

Thank you very much for your comments and suggestions. We improvised our Fig as suggested by reviewer.

Reviewer 2 Report

The revised manuscript « A sensitive impedimetric aptasensor based on carbon nanodots modified electrode for detection of 17ß-Estradiol” by Mat Zaid et al., still have to be improved because of the statistical error.

The limit detection of aptasensor was calculated using the 3σ/slope ratio (in which σ is the standard deviation of the mean Rct0) based on the curve plotted. This statistical method cannot be applied for a logarithmic curve (in Fig. 4).

To calculate LOD a linear curve should be used.

Author Response

Response: Thanks again and we really appreciated your comment. We agree that our logarithmic curve is statistically wrong due to negative curve relationship. therefore, a new logarithmic curve is based on equation bellow and makes it more senses.
impedance change % = Impedance probe – impedance E2 response/ impedance probe * 100%

Reviewer 3 Report

The manuscript has been improved after revisions according to the reviewers' comments. I recommend the paper to be published in the present form.

Author Response

We thank you for your careful reading of the manuscript and helpful comments and suggestions. We have made revisions according to your comments and suggestions, as described below

Reviewer comment: The manuscript has been improved after revisions according to the reviewers' comments. I recommend the paper to be published in the present form.

Response: Thanks again for your comment. We really appreciate the effort the reviewers have made in improving the quality of the articles

Round 3

Reviewer 2 Report

Thank you for your response.

Author Response

We are grateful to the editors and peer reviewers for their thoughtful comments that helped us improve the quality of our manuscript.